# GeoGramBench: Benchmarking the Geometric Program Reasoning in Modern LLMs

## Abstract

Geometric spatial reasoning forms the foundation of many applications in artificial intelligence, yet the ability of large language models (LLMs) to operate over geometric spatial information expressed in procedural code remains underexplored. In this paper, we address this gap by formalizing the `Program-to-Geometry` task, which challenges models to translate programmatic drawing code into accurate and abstract geometric reasoning. To evaluate this capability, we present **GeoGram-Bench**, a benchmark of 500 carefully refined problems organized by a tailored three-level taxonomy that considers geometric complexity rather than traditional mathematical reasoning complexity. Our comprehensive evaluation of 17 frontier LLMs reveals consistent and pronounced deficiencies: even the most advanced models achieve less than 50% accuracy at the highest abstraction level. These results highlight the unique challenges posed by program-driven spatial reasoning and establish GeoGramBench as a valuable resource for advancing research in symbolic-to-spatial geometric reasoning. Project page: https://github.com/LiAuto-DSR/GeoGramBench.

## 1 Introduction

Spatial reasoning is fundamental to both human cognition and artificial intelligence, supporting applications ranging from robotics and autonomous navigation to automated design [3]. With the rise of large language models (LLMs), interest has grown in evaluating their ability to interpret geometric transformations and spatial relations in complex environments [30, 26].

Mathematical geometric spatial reasoning is a specialized subdomain of spatial reasoning, requiring models to comprehend intricate geometric relationships and perform deep spatial reasoning. Researchers have recently developed multiple benchmarks including Mathverse [34], GeoSense [29], and Euclid [33] to assess LLMs' capabilities in visual geometry comprehension. Another emerging direction leverages procedural geometric code, such as Asymptote code, as a symbolic and structured interface for expressing geometry problems and probing spatial reasoning. While some existing benchmarks (e.g., AIME24 [21], MATH-500 [34]) include subsets containing Asymptote code, there is a lack of systematic, dedicated benchmarks specifically designed to evaluate LLMs' ability to perform program-driven spatial geometric reasoning. In this work, we formalize this unique setting as the `Program-to-Geometry` task, referring to the translation and abstraction process from procedural code to internal spatial representations.

Preliminary studies [22] have shown that current LLMs struggle to bridge procedural geometry code to spatial reasoning. We expanded these investigations on a broader range of models further corroborate these observations, confirming this pronounced deficiency. For example, as shown in Figure 1, advanced models such as DeepSeek-R1 [6] suffer substantial drops in accuracy—23.5% in AIME24 and 10.9% in MATH-500—when transitioning from text-only problems ($\mathbb{P}_T$) to those

with embedded procedural code ($\mathbb{P}_{TC}$). Similar trends are observed for models such as GPT-o1 [11] and QwQ-32B [28], collectively indicating critical limitations in their ability to construct reliable spatial representations from symbolic code. Furthermore, recent work [1] has highlighted the need to explore `Program-to-Geometry` spatial abstraction as a promising and under-investigated research direction.

Motivated by these findings, we introduce **GeoGramBench**, a dataset of 500 curated problems incorporating programmatic drawing code, designed to systematically assess both spatial-geometric abstraction capabilities and mathematical reasoning in LLMs. Our proposed taxonomy organizes problems into three categories—*Primitive Recognition*, *Local Relation Composition*, and *Global Abstract Integration*—based on the geometric complexity encoded in procedural code rather than traditional reasoning difficulty. Evaluation of 17 frontier LLMs reveals that even reasoning-oriented models (such as GPT-o1) achieve less than 50% accuracy on the most challenging level, underscoring the unique difficulty of this task and the urgent need for advances in spatial-reasoning model design.

This work makes the following contributions:

- We formalize the `Program-to-Geometry` translation task as a critical and underexplored capability for LLMs, encompassing not only the interpretation of procedural drawing code but also the downstream geometric reasoning it enables.

- We present **GeoGramBench**, a rigorously curated benchmark of 500 geometry problems with explicit procedural code, organized by a three-level taxonomy that enables comprehensive and fine-grained assessment of `Program-to-Geometry` competence.

- We conduct an extensive evaluation of 17 models, providing accuracy metrics and detailed behavior analyses aligned with our research questions. Our results highlight persistent weaknesses in geometric program reasoning, establishing GeoGramBench as a novel evaluation axis and fostering future advancements in spatially-grounded, symbolically-rich model training and analysis.

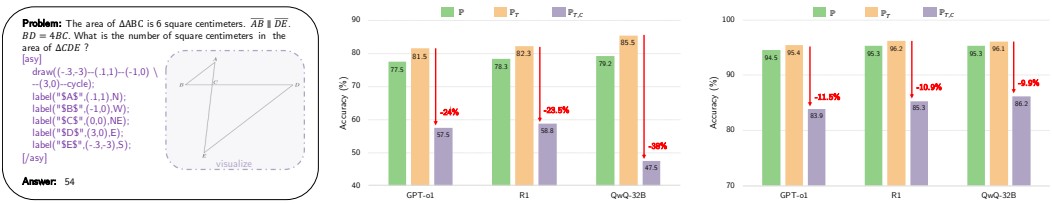

(a) Example of a problem from $\mathbb{P}_{TC}$ in MATH-500.

(b) Accuracy comparison of models on $\mathbb{P}_T$ vs. $\mathbb{P}_{TC}$ in AIME24.

(c) Accuracy comparison of models on $\mathbb{P}_T$ vs. $\mathbb{P}_{TC}$ in MATH-500.

Figure 1: Overview and performance analysis on text-only ($\mathbb{P}_T$) and text+code ($\mathbb{P}_{TC}$) geometry problems. (a) The procedural code is wrapped with `[asy][/asy]` and its geometric figure is visualized to facilitate understanding. (b) and (c) show accuracy comparisons of models on $\mathbb{P}_T$ and $\mathbb{P}_{TC}$ subsets in AIME24 ($|\mathbb{P}_{TC}| = 5$, $|\mathbb{P}_T| = 25$) and MATH-500 ($|\mathbb{P}_{TC}| = 42$, $|\mathbb{P}_T| = 458$), respectively. In both benchmarks, accuracy consistently drops for problems with procedural code.

## 2  Related Works

**Visual Geometric Perception**  To study visual geometric reasoning, several benchmarks such as Euclid [33], MM-Math [25], GeoSense [29], MathVerse [34], and MathVista [19] have been introduced, each incorporating visual geometric content. These datasets measure large multi-modal models' comprehension of visual geometric concepts and their handling of mathematical problems with visual components. Their focus is mainly on diagram interpretation rather than procedural geometric code understanding, which represents a different but equally important aspect of geometric spatial reasoning.

**Mathematical Reasoning Benchmarks**  A diverse array of benchmarks has been developed to evaluate the mathematical reasoning abilities of large language models (LLMs). Datasets such as GSM8K [2], MATH-500 [16], OlympiadBench [7], Minerva-MATH [13], CollegeMath [27],

MMLU-STEM [8], and AIME24 [21] primarily focus on algebraic, arithmetic, and word-problem reasoning. Many of these benchmarks target complex multi-step solutions, ranging from advanced high school mathematics to the level of international mathematical olympiads.

# 3 Program-to-Geometry

## 3.1 Task Definition

We define `Program-to-Geometry` as the task in which a model interprets procedural code to construct mathematical geometric representations, and subsequently reasons over these representations to solve geometry problems. This paradigm provides a comprehensive assessment of two fundamental capabilities: (a) the ability to accurately construct mathematical geometric diagrams from symbolic instructions, and (b) the ability to perform spatial reasoning and mathematical problem solving based on these constructed diagrams.

## 3.2 Taxonomy

Taxonomies for problem categorization are widely used across various fields, often focusing on dimensions such as topological complexity [35], logical intricacy [17], or the extent of required reasoning complexity (e.g., high school, graduate, olympiad-level) [21, 24, 9]. The `Program-to-Geometry` task fundamentally differs from these settings: it specifically examines the ability to map geometric code representations to geometric diagram understanding. Our preliminary analyses reveal that existing categorization schemes fail to capture the unique aspects and challenges of this space. Consequently, we propose a tailored taxonomy that better reflects the core competencies required for `Program-to-Geometry` translation.

As shown in Figure 2, based on reasoning difficulty annotations provided by the MATH-500 [16] dataset, difficulty is similarly distributed between text-only and text+code geometry problems, yet model performance diverges sharply. For instance, models like QwQ-32B perform worse on the easiest text+code problems than on the hardest, suggesting that reasoning complexity alone is not the determining factor.

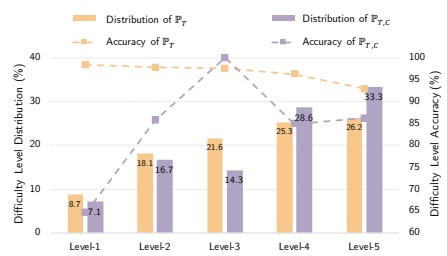

Instead, we propose a taxonomy whose primary principle is the construction of increasingly complex mathematical geometric diagrams from code. Our three-level categories are determined chiefly by the types and number of geometric elements involved, while also reflecting the depth of spatial reasoning required for each problem (see Figure 3):

Figure 2: Distribution of problem difficulty levels and QwQ-32B accuracy for text-only ($\mathbb{P}_T$) vs. text+code ($\mathbb{P}_{TC}$) geometry problems on MATH-500.

- *Primitive Recognition:* Problems involving procedural code that specify only one or two geometric primitives (e.g., points, lines, arcs, circles, polygons), focusing on basic mathematical properties such as length, area, or angle.

- *Local Relation Composition:* Problems with multiple local geometric elements, requiring the recognition, integration, and composition of spatial relationships among subcomponents of the diagram.

- *Global Abstract Integration:* Items demanding spatial direction, parameterization, recursion, 3D objects, composite structures, or advanced geometric operations (e.g., rotation, folding, projection), thus requiring not only the construction of complex diagrams but also global and stepwise spatial reasoning across the entire configuration.

## 3.3 Research Questions

Based on this task definition and taxonomy, we articulate the following research questions to structure our analysis of LLMs behavior in the `Program-to-Geometry` context:

**RQ1:**  *Is there evidence that LLMs can understand and represent basic geometric elements from program code?*

**RQ2:**  *How effectively can LLMs compose and abstract geometric elements into coherent spatial configurations as specified by program code?*

**RQ3:**  *How does chain-of-thought (CoT) reasoning influence LLMs' spatial geometric reasoning abilities with program code?*

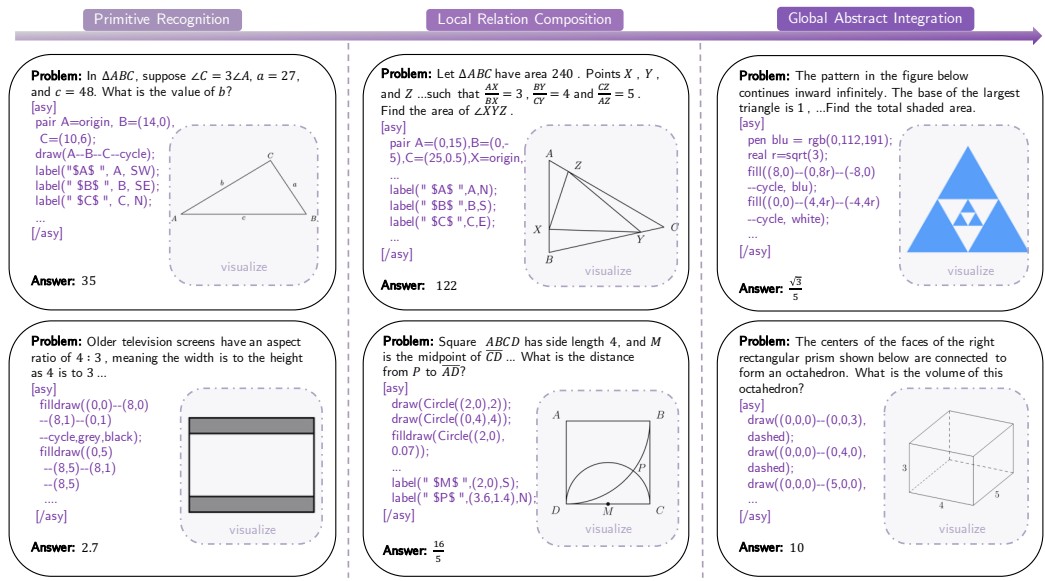

Figure 3:  Representative examples from GeoGramBench illustrating the three ascending `Program-to-Geometry` difficulty levels: *Primitive Recognition*, *Local Relation Composition*, and *Global Abstract Integration*. Each category is exemplified by two sampled problems, highlighting the increasing spatial complexity and abstraction across levels.

# 4   Benchmark Construction

In this section, we present the systematic construction process of **GeoGramBench**, a dedicated benchmark for `Program-to-Geometry` reasoning. We first introduce a critical challenge inherent to this task domain—answer leakage—before detailing our comprehensive data construction pipeline that forms the foundation of our benchmark (more details in Appendix D).

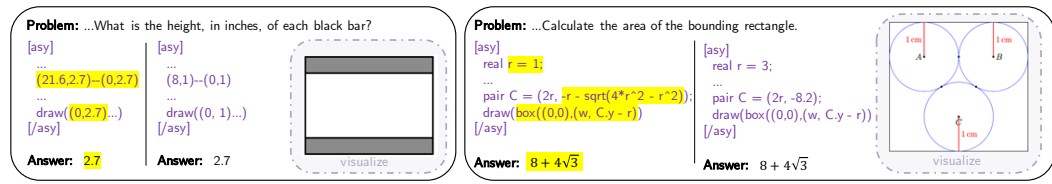

Figure 4: Illustration of two types of answer leakage in procedural code (highlighted in yellow): Left—**Direct leakage**, where the answer is explicitly given by a coordinate value in the Asymptote code (here, we rescale the coordinates to preserve the geometric shape); Right—**Indirect leakage**, where the answer can be computed from code parameters (in this case, we modify the procedural code to mask such critical information).

## 4.1 Answer Leakage Challenges

In the `Program-to-Geometry` task, a significant challenge arises from the potential for answer leakage within the code itself. The program code that generates geometric figures often contains precise numerical specifications that directly or indirectly reveal the answers sought. Benchmark like Math-500 [16], we discovered numerous instances where answers were directly embedded in the Asymptote code. Similar issues persist across various open-source geometry problem collections we collected. As illustrated in Figure 4, we categorize two types of answer leakage in the procedural code. **Direct leakage** occurs when the answer is explicitly encoded as a coordinate value in the Asymptote code (e.g., a circle's radius or segment's length). **Indirect leakage** occurs when the answer can be computed from code parameters or formulas.

## 4.2 Collection and Preprocessing

We first aggregated approximately 905K candidate problems from three open-source mathematics datasets, including NuminaMath-1.5 [15], HARP [32], and Omni-MATH [4], with a focus on sources rich in geometry content. We filtered for problems containing embedded Asymptote code by searching for `[asy]` and `[/asy]` tags, resulting in a subset comprising about 1% (9,260 problems). We then deduplicated this subset using an $n$-gram ($n = 8$) similarity approach [22], reducing the set to 1,782 unique items. Finally, by following the schema from s1 [22] and leveraging GPT-4o [10] for prompt-based classification, we selected only geometry problems, yielding 1,247 geometry-focused items for subsequent curation.

## 4.3 Human Refinement and Verification

To ensure data quality and suitability for geometry code understanding tasks, we implemented a two-stage manual verification process, conducted by a team of four experts (each holding a master's degree or higher in mathematics or related fields). The first round aimed to standardize problem types and formats, while the second round focused on enhancing overall problem quality.

In the **first round**, we performed initial screening and format normalization: (a) non-relevant questions (such as hyperlink chains, multi-part items, and proofs) were filtered out according to best practices from BigMath [1]; (b) convertible multiple-choice questions were transformed into open-form computation problems by removing options, while those not amenable to conversion were discarded entirely; and (c) answers were standardized into consistent LaTeX format. At the end of this screening, 547 candidate problems remained.

In the **second round**, we implemented a rigorous three-pronged refinement process to improve problem quality:

- Decontamination: To minimize community-sourced contamination, we systematically revised problem statements by removing redundant descriptive information that might enable direct textual inference. Additionally, we adjusted problem conditions and modified corresponding answers to maintain mathematical consistency. Furthermore, we adjusted the answer requirements (such as replacing queries about lengths with those about area, volume, or ratios) to further reduce the risk of leakage and promote authentic geometric reasoning.

- Answer Leakage Prevention: As detailed in Section 4.1, to address this task-specific vulnerability, we implemented two targeted strategies: systematically rescaling coordinates while preserving geometric relationships for direct leakage, and modifying or masking code parameters for indirect leakage. These interventions ensure that answers cannot be derived through mere code inspection (see Figure 4).

- Accuracy Verification: Each answer was manually checked for correctness; items with ambiguous, unverifiable, or doubtful solutions were removed.

Through this thorough process, we ultimately obtained 392 high-quality, contamination-free geometry problems for augmentation and evaluation.

## 4.4   Benchmark Augmentation

To enhance difficulty balance and problem diversity, we supplemented GeoGramBench with additional items: 5 geometry problems from AIME24 [21], 42 from MATH-500 [16], and 61 geometric problems adapted from Mathverse [34]. For the Mathverse subset, we selected representative solid geometry problems and manually transcribed diagrams into `matplotlib` code to diversify the procedural drawing code within the dataset. Our experiments indicate minimal impact from the choice of drawing language (see Appendix A). Altogether, GeoGramBench comprises 500 geometry problems, supporting robust evaluation across a variety of geometric phenomena.

## 4.5   Difficulty Categorization

Building on our theoretical and empirical insights in Section 3.2, we categorize all 500 GeoGram-Bench problems into three ascending difficulty levels—*Primitive Recognition*, *Local Relation Composition*, and *Global Abstract Integration*—based on the type and number of geometric elements and the spatial relationships involved (see Figure 3). The categorization is implemented through a combination of GPT-4o [10] assisted classification and thorough human expert review. The final distribution comprises 102, 279, and 119 problems for each category, respectively. GeoGramBench thus stands as the largest and most diverse `Program-to-Geometry` benchmark to date, establishing a rigorous testbed for spatially grounded language model evaluation.

# 5   Experiment

We benchmark 17 popular LLMs on GeoGramBench, providing a broad comparative analysis in this section. Section 5.1 details our evaluation framework and prompt engineering strategies. Section 5.2 introduces the tested models, followed by quantitative comparisons in Section 5.3.

## 5.1   Evaluation Protocols

For open-source models, we adopt the  [20] framework for evaluation, while for closed-source models, we utilize official APIs with identical prompt templates(*Let's think step by step and output the final answer within \boxed{}.*). All result parsing is standardized using  [20], with assistance from GPT-4o when necessary. Each problem is evaluated in a zero-shot setting: the model input consists strictly of the problem text and the procedural geometry drawing code. For each problem instance, we sample 8 responses using temperature 0.6, and report final accuracy as the mean over these 8 outputs, which balances model stochasticity and answer reliability.

## 5.2   Evaluation Models

We evaluate a total of 17 mainstream LLMs, including both proprietary APIs and leading open-source systems. The closed-source models include GPT-4o [10], GPT-o3-mini [23], the GPT-o1 series [11], and Gemini-Pro-1.5 [5]. The open-source models cover a wide range of scales, including DeepSeek-R1 [6], DeepSeek-v3-0324 [18], and QwQ-32B [28], as well as other prominent models from 32B down to 1.5B parameters: DeepSeek-R1-Distill variants [6], Bespoke-Stratos-32B [12], s1.1-32B [22], LIMO-32B [31], Sky-T1-mini-7B [14], and DeepScaleR-1.5B-preview [20].

## 5.3   Main Results

As shown in Table 1, all tested LLMs perform strongly on the *Primitive Recognition*, but accuracy drops steadily as geometric complexity increases. This downward trend is evident at the *Local Relation Composition* level and becomes most pronounced on *Global Abstract Integration*, where the highest accuracy is only 43.35% across all models.

Although GPT-o1 achieves a similar accuracy (86.76%) to DeepSeek-R1 (85.66%) in the *Primitive Recognition* category, a substantial performance gap emerges when evaluating *Global Abstract Integration*. GPT-o1 scores 43.35% in this more complex domain, whereas DeepSeek-R1 reaches only 40.38%. This significant difference may indicate a unique strength of closed-source models over open-source models in handling the most challenging tasks that require complex, abstract reasoning.

For open-source models, we observe a clear downward trend in performance as the model size decreases from 32B to 7B parameters. At the 32B scale, the reasoning-oriented QwQ-32B model achieves state-of-the-art results. However, academic models of similar scale, such as s1.1-32B and LIMO-32B, still exhibit a notable gap in performance compared to QwQ-32B.

| Model | Primitive | Compositional | Abstract | ALL |
|-------|-----------|---------------|----------|-----|
| **Closed-source Models** | | | | |
| GPT-o3-mini | 84.33 | 75.66 | 42.16 | 70.00 |
| GPT-o1 | **86.76** | **76.02** | **43.35** | **70.92** |
| GPT-o1-preview | 74.79 | 55.98 | 26.20 | 53.15 |
| GPT-o1-mini | 79.62 | 63.21 | 29.09 | 58.94 |
| GPT-4o | 39.81 | 21.29 | 4.96 | 21.40 |
| Gemini-Pro-1.5 | 49.26 | 31.79 | 15.92 | 31.64 |
| **Open-source Models** | | | | |
| DeepSeek-R1 | **85.66** | **75.27** | **40.38** | **69.17** |
| DeepSeek-v3-0324 | 80.57 | 68.89 | 27.67 | 62.05 |
| QwQ-32B | 85.17 | 73.12 | 37.92 | 67.20 |
| DeepSeek-R1-Distill-Qwen-32B | 79.78 | 67.83 | 35.92 | 62.68 |
| Bespoke-Stratos-32B | 62.50 | 42.56 | 17.02 | 40.55 |
| s1.1-32B | 75.37 | 58.96 | 26.58 | 54.60 |
| LIMO-32B | 76.59 | 59.63 | 25.53 | 54.98 |
| DeepSeek-R1-Distill-Qwen-7B | 72.79 | 58.74 | 24.16 | 53.38 |
| Sky-T1-mini-7B | 71.45 | 57.75 | 24.79 | 52.70 |
| DeepSeek-R1-Distill-Qwen-1.5B | 60.29 | 39.02 | 11.03 | 36.70 |
| DeepScaleR-1.5B-preview | 65.44 | 47.89 | 15.76 | 43.83 |

Table 1: Primitive: *Primitive Recognition*, Compositional: *Local Relation Composition*, Abstract: *Global Abstract Integration*. Accuracy (%) of selected closed-source and open-source LLMs on GeoGramBench across three difficulty levels. All models show a marked drop in performance on Abstract tasks, with no model exceeding 50% accuracy at this level. The highest results in both Closed and Open sources models are shown in bold.

# 6 Behavior Analysis of LLMs

We address our RQs through both quantitatively and qualitatively analyses base on benchmarking results and detailed model responses.

**RQ1:** *Is there evidence that LLMs can understand and represent basic geometric elements from program code?*

RQ1 investigates the fundamental ability of LLMs to recognize basic geometry elements, which can be quantitatively measured by the evaluation results of Primitive Recognition. As shown in Table 1, most of the models achieve 60% accuracy on the *Primitive Recognition* level, suggesting that they can effectively parse and build basic geometric scenes from procedural codes. Qualitatively, some of the model responses explicitly reveal the capability to interpret and reconstruct geometric information. As shown in Figure 5, models frequently examine the procedural code for geometry understanding: "Now, looking at the Asymptote code", "Let me parse the Asymptote code a bit", and "maybe I should try to visualize this". They can also identify simple geometric relationships according to the procedural code. For example, "c is (2,0), so c/2 is (1,0). So the inner arc is between points a/2 and c/2", and "path inner = arc(d, a/2, c/2, CW);...path outer = arc(d, c, a, CCW);". These behavior demonstrate that LLMs are intent and capable to map procedural code into internal geometric structures. In conclusion, modern LLMs are able to construct basic geometric representations from procedural code.

**RQ2:** *How effectively can LLMs compose and abstract geometric elements into coherent spatial configurations as specified by program code?*

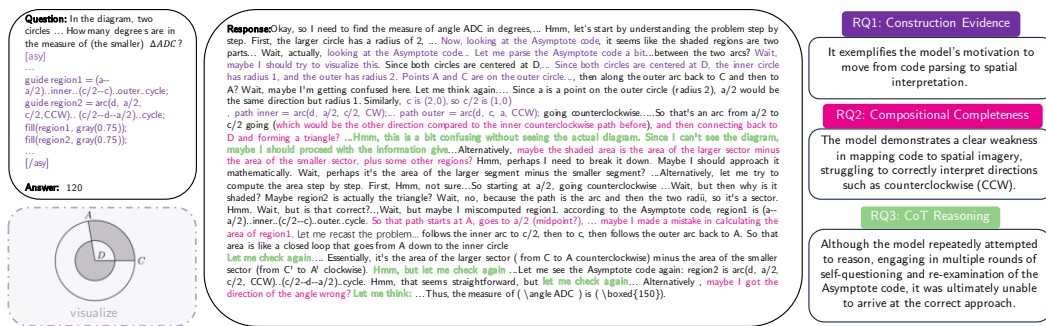

Figure 5: Illustrative solution process generated by the QwQ-32B model on a *Local Relation Composition* problem. The model initially attempts to construct spatial representations from the provided code, then interprets geometric elements such as direction and region, exhibiting behavior aligned with all three research questions (RQ1–RQ3): local construction, compositional integration, and chain-of-thought-based refinement. Multiple rounds of reflection and verification are observed, although these iterative steps do not consistently yield correct or fully integrated solutions.

RQ2 investigates LLMs' capability of the geometry composition and global representation abstraction. According to the results in Table 1, all models experience a significant drop in accuracy from *Compositional* problems to *Global Abstract Integration*. For example, GPT-o1 drops from 76.02% to 43.35%, and DeepSeek-R1 drops from 75.27% to 40.38%. These results indicate that current LLMs may lack of compositional and spatial abstraction ability to solve complex geometry problems. Qualitatively, while models can often parse and assemble some local structures, small errors in local constructions frequently appear, preventing LLMs to construct a complete and coherent global representation. As illustrated in Figure 5, a model may read a piece of code like "path inner = arc(d, a/2, c/2, CW)" and reason about directions ("which would be the other direction compared to the inner counterclockwise path before"), but a single mistake in local spatial assignment may generate downstream confusion: "maybe I got the direction of the angle wrong?... the actual angle between the points is \theta, so the area calculations still hold.". This phenomena suggests that modern LLMs may not good at capturing complex compositional geometry relationships for high level spatial reasoning. In summary, although LLMs have made progress in local geometric parsing, their ability to synthesize and reason over globally consistent spatial structures in Program-to-Geometry tasks remains limited.

**RQ3:** *How does chain-of-thought (CoT) reasoning influence LLMs' spatial geometric reasoning abilities with program code?*

Quantitatively, we observe a clear downward trend in accuracy as structural complexity rises. Since our benchmark taxonomy is based on geometric complexity rather than reasoning steps, this result suggests that most of the LLMs have difficulty in solving mathematical geometry problems with CoT. Qualitatively, while models frequently perform iterative self-reflection and verification of code ("Let me check again"), and repeatedly parse diagram instructions, their CoT trajectories rarely correct or update internal geometric understanding. For instance, the model may cycle through algebraic steps and verbalize uncertainty ("Hmm, this is a bit confusing without seeing the actual diagram. Since I can't see the diagram, maybe I should proceed with the information given."), yet consistently fails to resolve spatial relationships or integrate local shapes into a whole. This observation illustrates that CoT may lead LLMs fall into repetitive symbolic reasoning. Such repetitiveness does not beneficial for LLMs to construct high level spatial representations as a whole, even leading to confusion about complex geometry relationships. Although CoT improves LLM in mathematical reasoning, its ability to drive and update internal geometry understanding in complex spatial tasks remains fundamentally limited.

# 7   Discussion

**A Hypothesis on Internal Geometric Representations in LLMs**

Drawing on both quantitative results and behavior analyses, we hypothesize that large language models confronted with procedural geometry code engage in a multi-stage internal reasoning process closely aligned with the pipeline illustrated in Figure 6.

The process begins with the extraction of local geometric features or substructures ($\{z_1, z_2, \dots\}$) from the input text and code ($\{T, C\}$), corresponding to the abilities probed in RQ1. Our evidence shows that models are generally able to parse and represent these local primitives with high accuracy in simpler cases.

The next critical stage involves integrating these local elements into a coherent, global representation ($Z^1$), reflecting the compositional reasoning explored in RQ2. This is where we observe a pronounced bottleneck: small errors or ambiguities in local geometry can disrupt subsequent steps, making it difficult for models to build a structurally correct and complete diagram as complexity increases.

Subsequently, models iteratively attempt to update and refine their global geometric understanding, often through chain-of-thought (CoT) reasoning or self-reflective steps, in hopes of reconciling inconsistencies and clarifying spatial relationships. Despite such iterative efforts, our analysis of model outputs indicates that most fail to achieve robust global integration, as highlighted by the continued drop in accuracy and recurring spatial confusion on the most complex tasks (RQ3).

Finally, the model produces an answer ($A$), leveraging whatever spatial structure has been successfully constructed and refined. Our overall findings suggest that while LLMs can recognize and extract local geometric information, and to some extent initiate the integration process, there remain significant limitations in aggregating and refining these components into a globally consistent geometric representation for accurate problem solving. Overcoming these integration and synthesis difficulties is likely to be a key research frontier for closing the gap in `Program-to-Geometry` spatial reasoning.

These findings point to the need for future research on more robust scene composition and iterative spatial integration mechanisms in LLMs, as well as the development of benchmarks and training strategies tailored to these specific bottlenecks.

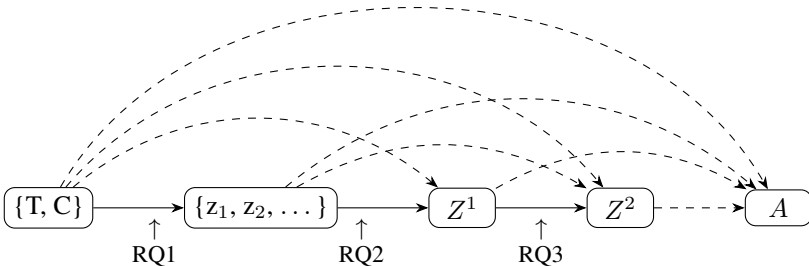

Figure 6: Illustration of the hypothesized multi-stage internal geometry representations process in LLMs for `Program-to-Geometry` tasks. The model first extracts local geometric substructures ($\{z_1, z_2, \dots\}$) from the problem statement ($\{T, C\}$), then integrates these into a coherent global structure ($Z^1$), which is further iteratively refined and updated ($Z^2, \dots$), before finally predicting the answer ($A$). Each stage corresponds to a core research question: RQ1 (local construction), RQ2 (compositional integration), and RQ3 (global abstraction and reasoning). Dashed arrows indicate how both input information and intermediate representations propagate throughout the process.

# 8 Conclusion

This work introduces the `Program-to-Geometry` task, which tests the capability of LLMs to map program code into geometric space, and **GeoGramBench** as a systematic benchmark for evaluating such geometric spatial reasoning abilities. Through a comprehensive analysis of 17 leading LLMs, we find that while models perform well on simple geometric constructions, their accuracy declines sharply for problems with higher geometric complexity—none surpassing 50% on the most advanced level. Our results highlight persistent challenges in complex geometric reasoning and emphasize the need for targeted advances in model design and training. GeoGramBench provides a robust foundation for future research on symbolic-to-geometric understanding in AI.

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

## A    Effect of Drawing Language on Program-to-Geometry Performance

A key motivation for our investigation is to determine to what extent challenges in `Program-to-Geometry` reasoning arise from the logic of geometric construction itself, rather than from surface-level code syntax or unfamiliarity with specific drawing languages. To test this, we translated 5 geometry questions containing Asymptote code from AIME24 and 42 questions from MATH-500 into equivalent Python `matplotlib` code, holding geometric content constant while varying only the programmatic language. As shown in Figure 7, QwQ-32B exhibits less than 1% difference in absolute accuracy between the Asymptote and Matplotlib versions on both benchmarks. This minimal gap provides strong evidence that the principal bottleneck in `Program-to-Geometry` task performance is not due to the choice of drawing language, but rather stems from deeper difficulties in spatial abstraction and geometric reasoning from code. This result reinforces our conclusion that surface syntax is not the main limiting factor for LLMs in this domain.

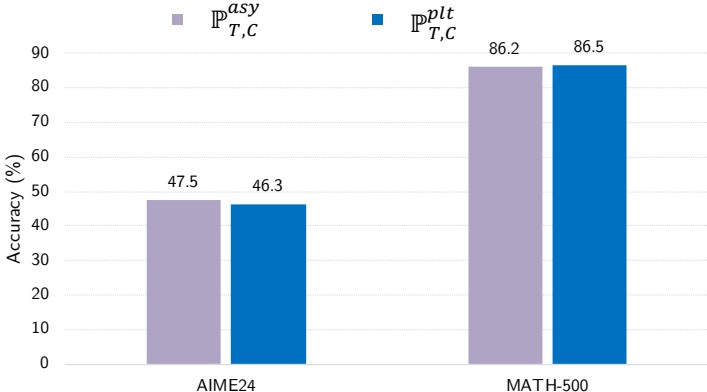

Figure 7: Comparison of QwQ-32B accuracy on equivalent geometry problems expressed in Asymptote versus Matplotlib code (AIME24 and MATH-500). The negligible performance gap demonstrates that `Program-to-Geometry` capability is independent of drawing language syntax.

**Example**

**Problem Statement:**
Rectangles $ABCD$ and $EFGH$ are drawn such that $D, E, C, F$ are collinear. Also, $A, D, H, G$ all lie on a circle. If $BC = 16$, $AB = 107$, $FG = 17$, and $EF = 184$, what is the length of $CE$?

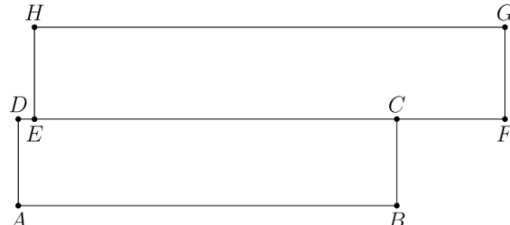

Figure 8: Visualization generated from the drawing code

**Drawing Code (Asymptote):**

```
import graph;
unitsize(0.1cm);
pair A = (0,0);
pair B = (70,0);
pair C = (70,16);
pair D = (0,16);
pair E = (3,16);
pair F = (90,16);
```

```
453     pair G = (90,33);
454     pair H = (3,33);
455     dot(A^^B^^C^^D^^E^^F^^G^^H);
456     label("\$A\$", A, S);
457     label("\$B\$", B, S);
458     label("\$C\$", C, N);
459     label("\$D\$", D, N);
460     label("\$E\$", E, S);
461     label("\$F\$", F, S);
462     label("\$G\$", G, N);
463     label("\$H\$", H, N);
464     draw(E--D--A--B--C--E--H--G--F--C);
```

**465 Drawing Code (Matplotlib):**

```
466     import matplotlib.pyplot as plt
467
468     A = (0, 0)
469     B = (70, 0)
470     C = (70, 16)
471     D = (0, 16)
472     E = (3, 16)
473     F = (90, 16)
474     G = (90, 33)
475     H = (3, 33)
476
477     for pt in [A, B, C, D, E, F, G, H]:
478     plt.plot(pt[0], pt[1], 'ko')
479
480     plt.text(A[0], A[1]-1, "\$A\$", ha='center', va='top', fontsize=13)
481     plt.text(B[0], B[1]-1, "\$B\$", ha='center', va='top', fontsize=13)
482     plt.text(C[0], C[1]+1, "\$C\$", ha='center', va='bottom', fontsize=13)
483     plt.text(D[0], D[1]+1, "\$D\$", ha='center', va='bottom', fontsize=13)
484     plt.text(E[0], E[1]-1, "\$E\$", ha='center', va='top', fontsize=13)
485     plt.text(F[0], F[1]-1, "\$F\$", ha='center', va='top', fontsize=13)
486     plt.text(G[0], G[1]+1, "\$G\$", ha='center', va='bottom', fontsize=13)
487     plt.text(H[0], H[1]+1, "\$H\$", ha='center', va='bottom', fontsize=13)
488
489     plt.plot([E[0], D[0], A[0], B[0], C[0], E[0]], [E[1], D[1], A[1], B[1],
490     C[1], E[1]], color='black')
491     plt.plot([E[0], H[0], G[0], F[0], C[0]], [E[1], H[1], G[1], F[1], C[1]],
492         color='black')
493
494     plt.xlim(-5, 95)
495     plt.ylim(-5, 38)
496     plt.gca().set_aspect('equal')
497
498     plt.axis('off')
499     plt.tight_layout()
500     plt.show()
```

## B   Taxonomy Classification Prompt Details

502  In constructing the GeoGramBench taxonomy, we categorized all 500 problems into three ascending
503  difficulty levels—*Primitive Recognition*, *Local Relation Composition*, and *Global Abstract Integra-*
504  *tion*—based primarily on the geometric and spatial complexity of each problem. This classification
505  process was conducted through a combination of large language model (GPT-4o) assisted clustering
506  and meticulous human expert correction. The initial clustering enabled an efficient, scalable filtering

of geometry problems, while human review ensured rigor, consistency, and alignment with the intended definitions of each difficulty level.

To ensure reproducibility and transparency, we provide below the actual LLM prompt used in the taxonomy assignment stage:

Given a geometry problem and its drawing code of diagram:
There are three categories of geometry problems:

1. Primitive Recognition
- The asy diagram/code contains very few geometric elements (e.g., one or two basic shapes, or minimal labeled points/lines).
- The solution can be reached with direct observation or a single basic calculation; no significant composition, auxiliary constructions, or synthesis are required.
- Tests only elementary recognition or reading from the diagram.
2. Local Relation Composition
- The asy diagram/code includes multiple geometric elements (points, lines, circles, polygons, etc.) combined in a finite and explicitly described way. - The solution requires synthesizing, coordinating, or combining several local relationships, auxiliary constructions, or properties. The process involves several steps, but remains within standard 2D geometry.
- The primary challenge is combining and reasoning locally among elements shown in the diagram.
3. Global Abstract Integration
- The asy diagram/code may be complex, recursive, or defined by folding, projection, 3D arrangement, or abstract/global spatial processes.
- The solution needs global synthesis: either full configuration analysis, recursive processes, or 3D/limit/extreme configuration reasoning.
- Tests the model's ability to reconstruct and reason about a highly integrated or abstract global geometric structure.

Instructions:
1. Classify the problem into one category: Primitive Recognition, Local Relation Composition, or Global Abstract Integration.
2. For geometric elements, consider only what is explicit in the asy code.
3. Judge the solution/reasoning requirement based on the problem's actual goal and what conceptual/computational effort is needed to reach the answer.
4. Briefly justify your classification: refer to relevant features in the diagram and in the problem's required reasoning process.
Output format:
- Category: [Primitive Recognition / Local Relation Composition / Global Abstract Integration]
- Justification: [A short explanation, citing relevant diagram elements and the level of reasoning/effort required.]

## C  Preventing Information Leakage in Procedural Geometry Code

A critical aspect of dataset curation for `Program-to-Geometry` evaluation is the prevention of information leakage through the procedural drawing code. In this context, information leakage refers to situations where the answer to a geometry problem is either explicitly or implicitly encoded in the program, enabling a model (or human) to bypass genuine geometric reasoning and instead extract the solution directly from code inspection.

We identify two primary forms of leakage:

- **Direct leakage**: The answer appears explicitly in the code, for example as a coordinate, length, or parameter value (e.g., a circle radius or segment described directly in the Asymptote code).

522    • **Indirect leakage**: The answer can be inferred by performing simple calculations or extract-
523      ing formula results from the parameters or structure of the code, even though it is not written
524      verbatim.

525  To mitigate these risks, we systematically reviewed all procedural code in the dataset. For direct
526  leakage, critical coordinates and parameters are rescaled or randomized while preserving the diagram's
527  structure. For indirect leakage, problem variables and code formulas are modified or masked to
528  preclude simple reverse engineering of the answer.

529  Below we present concrete examples comparing original and mitigated code for selected problems.
530  Each example includes its problem statement and paired Asymptote code, annotated as "before" and
531  "after" modification.

532  **Example 1:**

533  **Problem Statement:**
534  In $\triangle ABC$, point $F$ divides side $AC$ in the ratio $1 : 2$. Let $E$ be the point of intersection of side $BC$
535  and $AG$ where $G$ is the midpoint of $BF$. The length of $EC$ divided by the length of $BE$ is ?

536  **Answer: 3**

**Before modification (Leakage present):**

```
size(2.5inch);
pair A, B, C, E, F, G;
A = (0,3);
B = (-1,0);
C = (3,0);
E = (0,0);
F = (1,2);
G = intersectionpoint(B--F,A--E);
draw(A--B--C--cycle);
draw(A--E); draw(B--F);
label(\"$A$\",A,N);
label(\"$B$\",B,W);
label(\"$C$\",C,dir(0));
label(\"$E$\",E,S);
label(\"$F$\",F,NE);
label(\"$G$\",G,SE);
```

**After modification (Leakage mitigated):**

```
size(2.5inch);
pair A, B, C, E, F, G;
A = (0,3);
B = (-1,0);
C = (4,0);
E = (0,0);
F = (1.14, 2.14);
G = intersectionpoint(B--F,A--E);
draw(A--B--C--cycle);
draw(A--E); draw(B--F);
label(\"$A$\",A,N);
label(\"$B$\",B,W);
label(\"$C$\",C,dir(0));
label(\"$E$\",E,S);
label(\"$F$\",F,NE);
label(\"$G$\",G,SE);
```

Figure 9: Side-by-side comparison of Asymptote code: before (left) and after (right) information
leakage mitigation.

537  **Example 2:**

538  **Problem Statement:**
539  In rectangle $ABCD$, point $M$ is the midpoint of $\overline{AD}$. The area of $\triangle AMC$ is 12, and $\frac{AD}{AB} = \frac{3}{2}$. Find
540  the length of side $AD$.

541  **Answer: 8**

542

**Before modification (Leakage present):**

```
size(4cm);
draw((0,4)-(0,0)-(6,0)-(6,8)
-(0,8)-(0,4)-(6,8)-(0,0));
label(\"$A$\", (0,0), SW);
label(\"$B$\", (6, 0), SE);
label(\"$C$\", (6,8), NE);
label(\"$D$\", (0, 8), NW);
label(\"$M$\", (0, 4), W);
```

**After modification (Leakage mitigated):**

```
size(4cm);
draw((0,2)--(0,0)--(3,0)--(3,4)
--(0,4)--(0,2)--(3,4)--(0,0));
label("$A$", (0,0), SW);
label(\"$B$\", (3, 0), SE);
label(\"$C$\", (3,4), NE);
label(\"$D$\", (0, 4), NW);
label(\"$M$\", (0, 2), W);
```

Figure 10: Side-by-side comparison of Asymptote code: before (left) and after (right) information leakage mitigation.

## D  Detailed Benchmark Curation

We assemble a team of four experts (each holding a Master's degree or higher in mathematics or related fields) to ensure data quality. Our team manually verifies and refines samples from three aspects: question reformulation and standardization, decontamination, answer verification and leakage prevention.

### D.1  Question reformulation and answer standardization

**Question reformulation** The formulation of each sample in GeoGramBench should be simple QA pairs for convenient evaluation. To achieve this, we start to deal with multiple choice questions, proof-based questions and multi-part problems, which are not in QA format. Multiple choice questions can be transformed into open-ended computation problems by preserving the correct choice as the answer and removing all other choices. Some of the proof-based questions can be transformed into computation problems (like "Prove that $PA = 4PB$" can be rewrite to "Compute the ratio between $PA$ and $PB$"), whereas others are not suitable for such transformation (like "Prove that $AB \geq 3PR$). Multi-part problem always consists of several sub-problems, which can be simplified into a single question format by retaining one of the computable sub-questions. Questions amenable to conversion can be retained and reformulated into new QA samples, while others may be excluded from the benchmark. According to the aforementioned rules, our team members carefully assess the formulation of each question and perform corresponding modifications and deletion.

**Answer standardization** Considering the diversity and complexity of mathematical expressions, answer standardization is crucial for accurately evaluating model-generated responses. Our team manually modify the answer of each question by removing arithmetic operators (like $+, -$), letters and characters that irrelevant for computation and evaluation (like \text{cm^2}), and standardize each answer into LaTeX format as simple as possible (like simplify \frac{28}{\sqrt{7}}} to 4\sqrt{7}). The above operations successfully ensure the consistency of question formulation and answer standardization, which benefits subsequent data processing and contributes reliable benchmarking. The resulting subset contains 547 candidate samples.

### D.2  Decontamination

Most of the samples we collected originates from public datasets and internet resources, which indicates a high possibility that these data has already been included in the LLM's pre-training corpora. Besides, current data samples contains a certain degree of redundancy and unnecessary information, which may introduce unexpected bias to benchmarking. To mitigate the above influences as much as possible, our team manually perform data decontamination for all the 547 samples from three aspects:

**Extraneous information removal** We believe hyperlinks and code comments are not only unnecessary information for mathematic geometry spatial reasoning, but also introduce text bias for mathematic geometry problem reasoning. As a result, each member in our team carefully examine and delete all these contents in each question;

**Problem statement rephrasing** To prevent samples from being solved solely based on question statement, encourage LLM focus on mathematic geometry spatial reasoning, we reduce some comprehensive and specific mathematical expressions in question text. To minimize the overlap between LLM pre-training corpora and benchmarking samples, our team modifies the given condition and question objective of some samples;

**Coordinate modification** In some samples, the coordinates used to generate pictures are identical to the given conditions in the problem statement, which may enable LLM to derive answer through algebraic geometry reasoning based on text solely. Such problem solving approach cannot effectively evaluate the mathematic geometry spatial reasoning ability of LLM. To decrease the possibility of LLM using algebraic geometry problem solving approach, we adjust the coordinates in each samples program code, which maintains the geometric shape and relationship of the original picture. The above decontamination methods ensures each item in GeoGramBench is a completely new sample, contributing to valuable and reliable mathematic geometry spatial reasoning benchmarking.

### D.3 Answer Verification and Leakage Prevention

**Answer verification** We observe that some of the original answers are wrong to the corresponding questions after decontamination. To avoid such circumstances, we carefully verify the answer of each sample one by one by both referencing the original question from the Internet and calculate answer by ourself. The QA pairs that cannot be searched on the Internet are removed.

**Answer leakage prevention** We find some of the correct answers are already leaked in the code of samples during verification. As shown in Figure 9, 10, the answer can explicitly equals to the answer, or implicitly computed according to the code for generating image. This situation may allow LLM access the answer in advance, which harm to the evaluation of mathematic geometry spatial reasoning. To prevent answer leakage, our team manually revised the code for all samples once again by rescaling coordinates and masking codes with numbers. Answer verification and leakage prevention guarantee the correctness of all the samples and the fairness of benchmarking.

After human verification and refinement, we ultimately obtained 392 high-quality, contamination-free geometry problems for later augmentation and evaluation.

### D.4 Augmentation

We introduce additional samples to enhance difficulty and diversity of GeoGramBench: 5 geometry problems from AIME24 [21], 42 from MATH-500 [16], and 61 geometric problems adapted from Mathverse [34]. The 47 samples from AIME24 and MATH-500 are retained without modification dur to their high quality. For the Mathverse subset, we first filter 119 samples with two key words: Vision Intensive and Solid Geometry. These samples focus on solid geometry questions, with the majority of problem solving information presented in image. This advantages makes them highly suitable for mathematic geometry spatial reasoning evaluation. However, Mathverse only provides the original images without the plotting code for reproducing the picture. Thus, our team decide to write python matplotlib code with our own to construct new evaluation samples in GemGramBench. Notably, we do not ask for multimodal models (like GPT-4o) for help because such models performs poorly when transforming solid geometry picture to matplotlib code.

Altogether, GeoGramBench comprises 500 hand-crafted geometry problems, which contributes to valuable and reliable mathematic geometry spatial reasoning evaluation.

# E   More Behavior Analysis of LLMs

**Problem statement:**

In quadrilateral $ABCD$, angle $BAD$ and angle $CDA$ are trisected as shown. What is the degree measure of angle $AFD$?

**Answer: 80**

**Geometric Code:**

```
size(150);
pair A , B, C, D;
A = (0,0); B = (2, 4); C = (7,4); D = (7, -2);
draw( (0,0)--(2,4) -- (7,4) -- (7, -2)-- cycle);
label("$A$", A, SW);
label("$B$", B, NW);
label("$C$", C, NE);
label("$D$", D, SE);
pair E, F;
E = (4.5-.2,1-.2);
F = (5, 3);
draw(A--E--D);
draw(A--F--D);
label("$E$", E, N);
label("$F$", F, NW);
dot(A);dot(B);dot(C);dot(D);dot(E);dot(F);
label("$x$", (1, 1.5), S);
label("$x$", (2, 1), S+W);
label("$x$", (2, -1), N+N+N+W);
label("$y$", (5.5+.3, .5-.3), S); label("$y$", (6.5+.3, 0));
label("$y$", (5+.5, -1.5+.3));
label("$110^{\\circ}$",(2.5,3.5)); label("$100^{\\circ}$",(6.5-.2,3.5));
```

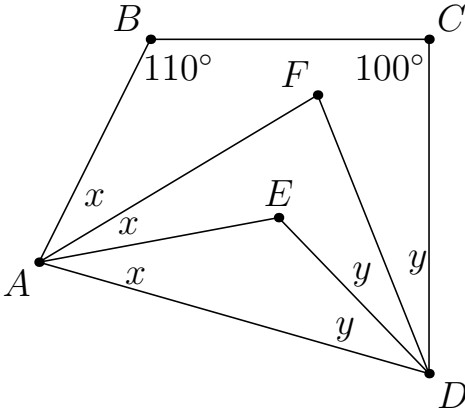

Figure 11: Visualization generated from the drawing code

\<think\> Okay, so I need to find the measure of angle AFD in this quadrilateral ABCD where angles BAD and CDA are trisected ...... First, let me note down the coordinates again to get a better idea: - A: (0,0) - B: (2,4) - C: (7,4) - D: (7,-2) . So quadrilateral ABCD has vertices at these points. Let me sketch this mentally. Starting at A (0,0), going to B (2,4), then to C (7,4), then to D (7,-2), and back to A. So AB is a line going up to the right, BC is a horizontal line to the right, CD goes down to the right, and DA goes straight left down to A. Wait, actually, D is at (7,-2), so DA would go from (7,-2) back to (0,0). Hmm, so DA is a line from the lower right back to the origin. Angles BAD and CDA are being trisected ...... Alternatively, since both E and F are connected from A to D, but through different paths? Maybe E is on one trisection and F on another? Hmm, this is getting a bit confusing without seeing the actual figure. Let me try to reconstruct based on coordinates. First, let's compute angles BAD and CDA to see their measures, which can then be trisected. Starting with angle BAD. This is the angle at vertex A between points B, A, and D. Coordinates: - Point B: (2,4) - Point A: (0,0) - Point D: (7,-2) So vectors AB and AD can be calculated ...... Now, arccos(0.1592). Let me use a calculator for better precision. Using a calculator, arccos(0.1592): cos(80 °) ≈ 0.1736 cos(81 °) ≈ 0.1564 So 0.1592 is between 80 and 81 degrees. Let me compute the difference: At 80 °: 0.1736 At 81 °: 0.1564 Difference between 80 and 81 ° is 1 °, and the difference in cos is -0.0172 per degree. We have 0.1592 - 0.1564 = 0.0028 above 81 °, so how much beyond 81 °? Wait, actually, since cos decreases as angle increases, 0.1592 is between 80 and 81 °, closer to 81 °. Let me do linear approximation: Let theta = 81 ° + d, where d is in degrees. cos(theta) ≈ cos(81 °) - (d * π/180) * sin(81 °) We have: 0.1592 ≈ 0.1564 - d * ( π/180)*0.9877 Wait, sin(81 °) ≈ 0.9877 So, 0.1592 - 0.1564 ≈ -d * ( π/180)*0.9877 0.0028 ≈ -d * 0.01745 * 0.9877 ≈ -d * 0.01725 Thus, d ≈ -0.0028 / 0.01725 ≈ -0.162 degrees So theta ≈ 81 ° - 0.162 ° ≈ 80.838 °, so approximately 80.8 °, so about 81 °, but very close to 81 °. Since the problem might want an exact value, perhaps 80 °, but maybe the answer is 80 °, considering possible rounding ...... Alternatively, maybe angle at B is 110 °, angle at C is 100 °, then angles at A and D would need to be 360 - 210 = 150 °, so angle BAD + angle CDA = 150 °, so if angle BAD is trisected into 3x and angle CDA into 3y, then 3x + 3y = 150 °, so x + y = 50 °, then angle AFD would be 180 - x - y = 130 °? Wait, no, in triangle AFD, angles at A and D would be x and y, so angle AFD = 180 - (x + y) = 180 -50=130 °, but that contradicts coordinate. Alternatively, maybe angle at B is 110 °, angle at C is 100 °, then angles at A and D would need to be 360 - 210 = 150 °, so angle BAD + angle CDA = 150 °, so if angle BAD is trisected into 3x and angle CDA into 3y, then 3x + 3y = 150 °, so x + y = 50 °, then angle AFD would be 180 - x - y = 130 °? Wait, no, in triangle AFD, angles at A and D would be x and y, so angle AFD = 180 - (x + y) = 180 -50=130 °, but that contradicts coordinate. Alternatively, maybe angle AFD is an external angle or something else ...... which is 80.8 °, so the closest integer is 81 °. Since the problem likely expects an integer, I'll go with $\boxed{81}$ as the answer. **Final Answer** $\boxed{81}$ To find the degree measure of angle $\angle AFD$ in quadrilateral $ABCD$ where angles $\angle BAD$ and $\angle CDA$ are trisected, we start by analyzing the given coordinates and using vector calculations. The coordinates of the points are: - $A = (0,0)$ - $B = (2,4)$ - $C = (7,4)$ - $D = (7,-2)$ - $F = (5,3)$ We need to determine the angle at point $F$ between points $A$, $F$, and $D$. This angle is formed by the vectors $\overrightarrow{FA}$ and $\overrightarrow{FD}$. 1. **Calculate vectors:** - $\overrightarrow{FA} = A - F = (0 - 5, 0 - 3) = (-5, -3)$ - $\overrightarrow{FD} = D - F = (7 - 5, -2 - 3) = (2, -5)$ 2. **Compute the dot product:**

$$\overrightarrow{FA} \cdot \overrightarrow{FD} = (-5)(2) + (-3)(-5) = -10 + 15 = 5$$

3. **Compute the magnitudes of the vectors:** - $|\overrightarrow{FA}| = \sqrt{(-5)^2 + (-3)^2} = \sqrt{25 + 9} = \sqrt{34}$ - $|\overrightarrow{FD}| = \sqrt{2^2 + (-5)^2} = \sqrt{4 + 25} = \sqrt{29}$ 4. **Calculate the cosine of the angle:**

$$\cos(\theta) = \frac{\overrightarrow{FA} \cdot \overrightarrow{FD}}{|\overrightarrow{FA}||\overrightarrow{FD}|} = \frac{5}{\sqrt{34}\sqrt{29}} = \frac{5}{\sqrt{986}}$$

5. **Determine the angle using the arccosine function:**

$$\theta = \arccos\left(\frac{5}{\sqrt{986}}\right)$$

Using a calculator, we find that:

$$\theta \approx 80.8°$$

Rounding to the nearest degree, the measure of angle $\angle AFD$ is approximately $81°$. Thus, the final answer is $\boxed{81}$.

650

**Problem Statement:** In the figure below, quadrilateral $CDEG$ is a square with $CD = 3$, and quadrilateral $BEFH$ is a rectangle. If $BE = 5$, how many units is $BH$? Express your answer as a mixed number.

**Answer:** $1\frac{4}{5}$

**Geometric Code:**

```
unitsize(5mm);
defaultpen(linewidth(.7pt)+fontsize(8pt));
pair A=(0,0), B=(3,0), C=(6,0), D=(9,0), Ep=(9,3), G=(6,3);
pair F0=bisectorpoint(B,2*Ep-B), H0=bisectorpoint(Ep,2*B-Ep);
pair H=extension(B,H0,A,G);
pair F=extension(Ep,F0,A,G);
draw(H--B--Ep--F--A--D--Ep--G--C);
label("$A$",A,S);
label("$B$",B,S);
label("$C$",C,S);
label("$D$",D,S);
label("$E$",Ep,E);
label("$F$",F,N);
label("$G$",G,NW);
label("$H$",H,NW);
```

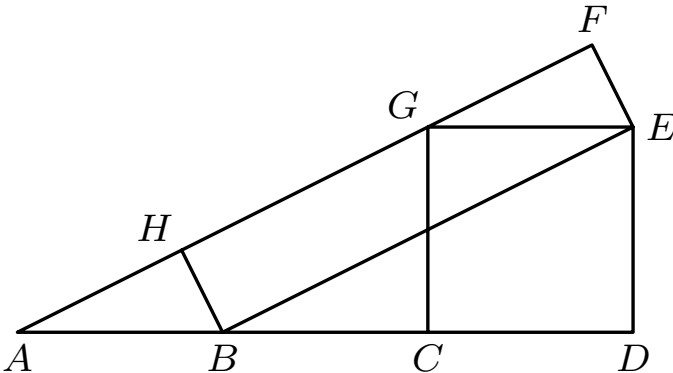

Figure 12: Visualization generated from the drawing code

Okay, let's see. I need to find the length of BH in the figure provided. The problem states that CDEG is a square with CD = 3, so each side of the square must be 3 units long. Also, BEFH is a rectangle, and BE is given as 5 units. Hmm, I should start by trying to understand the coordinates of the points based on the Asymptote code given.

First, looking at the Asymptote code, they define point A at (0,0), B at (3,0), C at (6,0), D at (9,0), E is labeled as Ep which is (9,3), and G is at (6,3). So, let me note down these coordinates:
- A: (0, 0) - B: (3, 0) - C: (6, 0) - D: (9, 0) - E: (9, 3) - G: (6, 3)
Wait, so quadrilateral CDEG is a square ...... Wait, that would not form a square because E to G is left 3 units and G to C is down 3 units? Wait, actually, if E is (9,3) and G is (6,3), then EG is 3 units left, and then GC is from (6,3) to (6,0), which is 3 units down. So actually, CDEG is a square with sides of 3 units each. So that checks out.
Now, quadrilateral BEFH is a rectangle ...... Alternatively, maybe the square is oriented differently? But since the problem didn't specify orientation, but the Asymptote code had coordinates with C at (6,0), D at (9,0), E at (9,3), G at (6,3). So in that code, it's a square with sides aligned to the grid, so CD is horizontal, DE vertical, etc. So perhaps I should stick with that coordinate system but adjust BE's length ...... Wait, but maybe it's better to compute coordinates step by step. Alternatively, since BEFH is a rectangle, then H must be such that BH is equal and parallel to EF, and FH is equal and parallel to BE. Alternatively, since BE is from B to E, then to get to F from E, we move in the direction perpendicular to BE ...... Alternatively, perhaps I chose the direction incorrectly. Let me try with the other perpendicular direction. If instead of (3,4), we take (-3,-4). Then vector EF would be (-3,-4). Then point F would be E(3,3) + (-3,-4) = (0,-1). Then vector FH would be equal to BE? Wait, no. Wait, in a rectangle, opposite sides are equal. So FH should be equal and parallel to BE. Hmm, perhaps I need to think differently. Alternatively, since BE is one side, and EF is another side, then FH should be equal and parallel to BE, and HB should be equal and parallel to EF.
So if vector BE is (-4,3), then vector FH should also be (-4,3). Starting from F, adding vector BE gives H = F + BE. Wait, but maybe I need to think of the rectangle as B -> E -> F -> H -> B ...... In the Asymptote figure, points are: A(0,0), B(3,0), C(6,0), D(9,0), E(9,3), G(6,3) We need to find H's coordinates.H is extension(B, H0, A, G). Let's try to compute H0 and then the line. First, H0 is bisectorpoint(Ep, 2*B - Ep). Ep is E(9,3). 2*B is (6,0). So 2*B - Ep is (6-9,0-3)= (-3,-3). So H0 is the bisectorpoint between Ep(9,3) and (-3,-3). Assuming bisectorpoint is the midpoint: midpoint is ( (9-3)/2, (3-3)/2 ) = (3,0). Which is point B. So line BH0 is the line from B(3,0) to H0(3,0), which is a point. Not helpful. So this suggests the midpoint is not the correct interpretation. Alternatively, maybe bisectorpoint returns a point along the angle bisector. Suppose at point Ep(9,3), between the direction to B(3,0) and to (2*B - Ep)=(-3,-3). The angle at Ep between those two points. The angle bisector direction would be the direction that is the angle bisector between vectors Ep-B and Ep-(2*B - Ep). Vector Ep - B = (9-3, 3-0)=(6,3) Vector Ep - (2*B - Ep) = Ep - (-3,-3) = (9 - (-3), 3 - (-3)) = (12,6) So the two vectors are (6,3) and (12,6), which are in the same direction (since (12,6)=2*(6,3)). So the angle between them is zero, so the bisector is the same line. Hence, the angle bisector is undefined, or the line itself. So H0 is along that line.
Hence, the line BH0 would be the line from B(3,0) to along the direction of Ep-B, which is the line towards E. So the line BE itself ...... Ah, yes! Because vector BE is (6,3), and vector HB is (3/5, -6/5). Their dot product is indeed zero. So the calculation is correct. Therefore, the answer is $\boxed{1}$.

**Final Answer** $\boxed{1}$ Given quadrilateral (CDEG) is a square with (CD = 3) and quadrilateral (BEFH) is a rectangle with (BE = 5), we need to find the length of (BH).
1. **Coordinates and Setup**: - In the Asymptote code, points (C), (D), (E), and (G) form a square with (CD = 3). The coordinates are (C(6,0)), (D(9,0)), (E(9,3)), and (G(6,3)). - The problem states (BE = 5), but in the Asymptote coordinates, (BE) is calculated as ($\sqrt{(9-3)^2 + (3-0)^2} = \sqrt{45} = 3\sqrt{5}$). To scale this to (BE = 5), the scaling factor is ($\frac{5}{3\sqrt{5}} = \frac{\sqrt{5}}{3}$).
2. **Finding Coordinates of (H)**: - In the Asymptote figure, line (AG) is from (A(0,0)) to (G(6,3)). The parametric form of (AG) is ((6t, 3t)). - Point (F) on (AG) is found by ensuring (EF) is perpendicular to (BE). Solving, (F) is at (($\frac{42}{5}, \frac{21}{5}$)). - Point (H) is determined by the rectangle properties, resulting in (($\frac{12}{5}, \frac{6}{5}$)).
3. **Calculating (BH) in Asymptote Coordinates**: - Coordinates of (B(3,0)) and ($left(\frac{12}{5}, \frac{6}{5}$)). - Distance (BH) is ($\sqrt{\left(3 - \frac{12}{5}\right)^2 + \left(0 - \frac{6}{5}\right)^2} = \sqrt{\left(\frac{3}{5}\right)^2 + \left(-\frac{6}{5}\right)^2} = \frac{3\sqrt{5}}{5}$).
4. **Scaling to Problem's Coordinates**: - Scaling factor ($\frac{\sqrt{5}}{3}$) is applied to the Asymptote length of (BH), resulting in ($\frac{3\sqrt{5}}{5} \times \frac{sqrt5}{3} = 1$).
Thus, the length of (BH) is ($\boxed{1}$).

671

## F Limitation and Future Work

Although GeoGramBench currently focuses on procedural code in geometry, the framework and insights developed here may generalize to broader domains where procedural descriptions interact with spatial or relational reasoning. Our present analysis is largely empirical and focuses on observable model behavior, without providing deeper theoretical explanations for these shortcomings. In future work, we plan to conduct more in-depth investigations into the underlying causes of failures on `Program-to-Geometry` tasks using this dataset, and to explore reinforcement learning as well as other targeted training strategies to explicitly enhance spatial reasoning and abstraction in LLMs. We encourage further research to expand upon this benchmark, develop more sophisticated probing methods, and systematically explore model behaviors under diverse procedural spatial contexts, ultimately advancing a deeper understanding of spatial reasoning capabilities in large language models.

