# OpenReview forum: "GeoGramBench: Benchmarking the Geometric Program Reasoning in Modern LLMs"
_NeurIPS.cc/2025/Datasets_and_Benchmarks_Track — Submitted to NeurIPS 2025 Datasets and Benchmarks Track_

### Official Review · Reviewer_gGrj · 2025-07-01

**Rating:** 4
**Confidence:** 2

**Summary:**

This paper introduces GeoGramBench, a benchmark of 500 curated geometry problems designed to evaluate LLMs' ability to perform Program-to-Geometry reasoning—translating procedural drawing code into spatial understanding. The tasks span three difficulty levels, and 17 LLMs are benchmarked. Results show that while most models can handle basic shapes, performance drops significantly on complex reasoning, revealing critical limitations in symbolic-to-spatial abstraction.

**Dataset Code Accessibility:**

Yes

**Dataset Code Comments:**

it looks good to me.

**Ethical Considerations:**

No, there are no or only very minor ethics concerns

**Final Justification:**

I would like to maintain my positive score.

**Limitations Weaknesses:**

The reviewer is curious about task-specific models and large language models. I would like to know whether fine-tuning a large language model on this dataset would lead to better results.

**Strengths Contributions:**

1. The paper proposes a novel and underexplored task that bridges code and geometric reasoning.
2. The benchmark is carefully curated, with thorough quality control and answer leakage prevention.
3. The evaluation covers a wide range of LLMs and provides clear insights into model limitations and reasoning gaps.

---

> ### Author Rebuttal · Authors · 2025-07-29
>
> Thank you for your thoughtful review and for raising the question regarding the impact of fine-tuning. Our work is the first to propose the Program-to-Geometry task, which differs from VLM benchmarks by evaluating spatial reasoning without reliance on perception modules, thereby removing the confounding effects of visual perception. Instead, it adopts a formal and precise geometric representation in Euclidean space via procedural code. To address the lack of resources in this area, we introduce GeoGramBench, the first large-scale benchmark with 500 curated Program-to-Geometry problems—an order of magnitude more than prior datasets.
>
> Following your suggestions, we have (1) conducted additional supervised fine-tuning experiments and (2) performed ablation studies and analysis to evaluate the effectiveness of the data and to disentangle the effects of data and model capacity. All supplementary experiments and analyses will be included in the updated version of the paper. Should you have any further suggestions for improvement, we would be glad to incorporate them.
>
> > The reviewer is curious about task-specific models and large language models. I would like to know whether fine-tuning a large language model on this dataset would lead to better results.
>
> (Since this suggestion was also raised by Reviewers U3SF and NfAE, we provide the same response here.)
>
> We conducted data ablation experiments to systematically assess the impact of adding domain-specific examples.
> The baseline in our study is the s1.1-32B model [Muennighoff et al., 2025], which is based on Qwen2.5-32B-Instruct and obtained via supervised training on the s1k dataset—a collection of 1,000 general reasoning examples released to enhance long chain-of-thought reasoning capabilities.
>
> - Data: Due to the scarcity of high-quality programmatic geometry data in the community, we sampled 200 problems from GeoGramBench (spanning three complexity levels) as a test set, while the remaining 300 were used for training. These 300 training problems were further annotated with reasoning chains via DeepSeek-R1, forming a distillation dataset.
>
> - Experimental Setup: We performed ablation studies by incrementally adding 50, 100, 150, and 300 code-based geometry training samples from GeoGramBench to the s1k data, and training on the combined dataset. This approach allowed us to systematically observe the impact of increasing task-specific data exposure, with all added samples included in the training set along with the original s1k data.
>
> Experimental Result
> |Exp.|# Added Samples| Primitive| Compositional | Abstract | Avg.|$\Delta$|
> |:------:|:------------:|:----------:|:-----------------:|:---------:|:----:|:----:|
> |1|0|59.38|46.00|15.80|38.18|-|
> |2|50|57.99|49.61|16.18|39.80|+1.62|
> |3|100|61.22|49.66|18.62|41.20| +3.02|
> |4|150|61.11|50.90|18.38|41.73| +3.55|
> |5|300|61.11|51.42|17.83|41.79| +3.60|
>
> Experimental Analysis
> 1. Adding domain-specific data leads to a clear improvement in accuracy. Comparing Exp. 3 with Exp. 1, the average accuracy increases by 3.02 percentage points. This demonstrates that the model becomes more familiar with this type of data, effectively resolving many ambiguities in understanding geometric code as well as enhancing logical and representational reasoning over geometry. These results provide strong evidence for the data effectiveness of GeoGramBench in improving model performance.
> 2. Notably, while as few as 100 in-domain examples (Exp. 3) substantially enhance accuracy across all difficulty levels, further increasing the data (Exp. 5 with triple the added samples) yields only marginal gains (e.g., Exp. 5: 3.60 vs. Exp. 3: 3.02). The improvement quickly plateaus, indicating that the model’s performance is limited by its inherently weak geometric spatial representation abilities, which cannot be significantly improved simply by scaling up training data.
> 3. Even with domain-specific fine-tuning, the absolute performance remains unsatisfactory. For example, in Exp. 5, the highest average accuracy achieved is only 41.79%, and performance on the most challenging “Abstract” subset is especially poor (17.83%). This again highlights the intrinsic deficiency in symbolic-to-spatial reasoning within current models.
>
>
>
> In summary, although our experiments were limited to small data batches due to the scarcity of available resources, the results clearly indicate that while the unfamiliarity with task-specific data does contribute to low initial performance, the more fundamental limitation lies in the model’s intrinsic geometric spatial reasoning capability—a challenge that is difficult to overcome through additional data alone.

---

### Official Review · Reviewer_9Qhh · 2025-07-01

**Rating:** 4
**Confidence:** 3

**Summary:**

GeoGramBench introduces the Program‑to‑Geometry task, challenging large language models to translate Asymptote code into accurate internal geometric representations and solve the associated problems, and presents a 500‑question benchmark to evaluate this ability, which is organized into three levels (Primitive Recognition, Local Relation Composition, Global Abstract Integration). Extensive experiments on 17 leading LLMs reveal strong performance on simple shape recognition but a steep accuracy drop on globally integrated tasks, exposing a critical weakness in the current spatial‑composition skills of LLMs. By formalizing the task, providing the first leakage‑free dataset with code‑grounded geometry, and delivering quantitative and qualitative analyses of model behavior, the paper establishes a new axis for analyzing and improving symbol‑to‑space reasoning in future LLMs.

**Dataset Code Accessibility:**

Yes

**Ethical Considerations:**

No, there are no or only very minor ethics concerns

**Final Justification:**

1. Authors added a full subtype breakdown and failure taxonomy, now giving quantitative insight into where models fail. This directly satisfies the biggest technical gap in the original review.

2. Token‑Budget‑Forcing experiments show that varying CoT depth does not materially change the conclusions, bolstering the robustness of the results.

3. Benchmark still targets contest‑style Asymptote geometry, but the paper now justifies this choice and positions it as a first step.

4. External baselines for human, oracle, and tool‑aided are still absent, even though authors convincingly argue they are beyond the current scope and outline them as future work.

After rebuttal, the submission is technically solid, introduces a genuinely novel evaluation axis, provides a well‑curated resource, and now includes analysis to support its claims, but still with some limitations.

**Limitations Weaknesses:**

1. All seventeen models are evaluated only in a zero‑shot, chain‑of‑thought (CoT) setting. The paper does not report ablations without CoT prompts, alternative prompting strategies, or any tool‑augmented baselines. Consequently, the claim that CoT “fails to repair spatial misconceptions” is not based on a quantitative comparison.
2. GeoGramBench deliberately focuses on planar/solid contest problems expressed in Asymptote, excluding other spatial‑reasoning modalities such as everyday 3D scenes, topological task settings. This design choice limits the external validity of conclusions about spatial reasoning.
3. Table 1 lists only LLM accuracies. It would be better to compare trained geometry solvers that can execute the code, human solvers, and an oracle that sees the rendered diagram. Behavior Analysis of LLMs provides rich qualitative examples, but no aggregated statistics on which primitives, relations, or global operations dominate the error budget. Without such granularity, it is hard for future work to target the most impactful failure modes.

**Strengths Contributions:**

1. This paper establishes a novel evaluation horizon by formalizing the Program‑to‑Geometry task, which requires models to translate Asymptote code into internal geometric representations, and supports it with a thoughtfully motivated three‑tier taxonomy that discriminates between primitive, compositional, and global reasoning challenges, that not addressed by existing visual‑diagram or text‑based math benchmarks.
2. GeoGramBench, including 500 problems from multiple sources, has been through two rounds of expert review, answer leakage has been systematically eliminated, and all solutions are standardised in LaTeX, yielding a leakage‑free dataset that sets a new quality bar for symbol‑to‑space benchmarks.
3. The authors report a wide‑ranging empirical study over seventeen leading LLMs under a uniform chain‑of‑thought protocol; the universal collapse in accuracy to below fifty percent on the highest abstraction tier, coupled with the paper’s qualitative dissection of failure modes such as direction reversal and incomplete global synthesis, provides clear, actionable insight into the current limitations of large language models for spatial reasoning and offers a solid foundation for future research.

---

> ### Author Rebuttal · Authors · 2025-07-31
>
> Thank you for taking the time to provide such detailed and useful feedback. Based on your suggestions, we have included further clarifications and relevant experiments, and we would appreciate any additional feedback to help us further improve the work.
>
> > All seventeen models are evaluated only in a zero‑shot, chain‑of‑thought (CoT) setting. The paper does not report ablations without CoT prompts, alternative prompting strategies, or any tool‑augmented baselines. Consequently, the claim that CoT “fails to repair spatial misconceptions” is not based on a quantitative comparison.
>
> Thank you for this valuable suggestion. In response, we have added a quantitative experiment analyzing the effect of Token Budget Forcing (BF) [Muennighoff et al., 2025]—a technique that expands the model’s CoT by repeatedly appending "Wait" tokens (N-Ignore) whenever the model attempts to conclude its generation. This method has shown effectiveness in improving reasoning ability; for example, applying BF to the s1 model raised performance on AIME24 from 50.0 to 56.7. Therefore, we selected this technique to further evaluate the quantitative impact of extended reasoning on GeoGramBench.
>
> We conducted these experiments using the s1.1-32B model [Muennighoff et al., 2025], which is specifically designed for strong long CoT reasoning.
>
> |BF| N-Ignore|Token Count | Primitive | Compositional | Abstract | Avg.  |
> |:----:|:----:|:-----------:|:---------:|:-------------:|:--------:|:-----:|
> |No| 0|10544         | 75.37     | 58.96         | 26.58    | 54.60 |
> |Yes| 1 |11336       | 76.47     | 58.78         | 26.89    | 54.90 |
> |Yes| 2|12319       | 76.47     | 58.78         | 26.89    | 54.90 |
> |Yes| 4|15245       | 75.49     | 59.49         | 26.89    | 54.90 |
> |Yes| 6|18710       | 74.50     | 59.13         | 26.89    | 54.40 |
>
> While the BF approach extends the CoT and yields minor improvements on GeoGramBench, these gains are marginal. As the value of N-Ignore increases, the model's reasoning chains become significantly longer (as reflected by the Token Count), thereby expanding its reasoning capacity to some extent. However, these extended chains result in only slight performance gains, suggesting that simply generating longer reasoning chains is insufficient to overcome the underlying limitations in symbolic-to-spatial reasoning. In this work, we only tested the BF strategy; exploring additional reasoning strategies for more comprehensive quantitative evaluation remains an important direction for future work.
>
> > GeoGramBench deliberately focuses on planar/solid contest problems expressed in Asymptote, excluding other spatial‑reasoning modalities such as everyday 3D scenes, topological task settings. This design choice limits the external validity of conclusions about spatial reasoning.
>
> Everyday 3D scenes and topological tasks are indeed important application areas, but they fall outside the immediate scope of our current study. Given the complexity of spatial reasoning and the lack of general solutions for LLMs in broader contexts, we intentionally focus on geometric representations. Geometric figure reasoning is a domain where clear, accurate, and structured descriptions are possible, making it well suited for systematic analysis. Additionally, we chose to use programmatic drawing code because it provides a formalized representation of geometric figures, free from the visual and perceptual attributes inherent to images. This allows us to investigate the fundamental and essential aspects of spatial reasoning in a more controlled setting.
>
>
>
> > Table 1 lists only LLM accuracies. It would be better to compare trained geometry solvers that can execute the code, human solvers, and an oracle that sees the rendered diagram.
>
> Thank you for the constructive suggestion. We would like to clarify why the proposed comparisons are outside the main scope of our defined task objectives:
>
> 1. Code Executors & Diagram Oracles Misalign with Core Task Design: Program-to-Geometry is designed as a foundational task for evaluating the intrinsic geometric spatial reasoning abilities of LLMs directly from code. The core principles are: (a) eliminating perceptual factors (unlike VLM tasks such as MathVerse/MathVista), (b) focusing on abstract geometric properties and symbolic inference, and (c) targeting the internal visualization capability. Comparing to code executors (which do not perform reasoning) or diagram oracles (which rely on direct visual input) would fundamentally shift the task away from its intended focus.
>
> 2. Human Benchmark&mdash;Practical Challenges: Conducting a large-scale, rigorous human evaluation is especially challenging, as it would require experts who are both skilled in solving geometry problems and able to accurately understand procedural code like Asymptote—a rare skill set among human participants. As GeoGramBench’s key goal is to establish the first large-scale LLM benchmark for this domain, we prioritized reporting LLM performance baselines. Comprehensive human studies are an important direction for future work.
>
> > Behavior Analysis of LLMs provides rich qualitative examples, but no aggregated statistics on which primitives, relations, or global operations dominate the error budget. Without such granularity, it is hard for future work to target the most impactful failure modes.
>
> Thank you for the insightful suggestion. We agree that providing more fine-grained error type analysis would increase the value of our benchmark and facilitate targeted research advancement. A detailed classification based on specific geometric primitives, their combinations, and abstract operations could indeed enable more precise behavioral analysis of LLMs. However, constructing such an exhaustive taxonomy is nontrivial because the number of possible interactions and composition patterns across dimensions (primitives, relations, abstraction levels) is combinatorially large and highly context-dependent.
>
> In response to your suggestion, we conducted additional experiments to provide finer error categorization aligned with problem-solving objectives. Each of the three complexity levels—Primitive Recognition, Local Relation Composition, and Global Abstract Integration—was subdivided according to subtask types including length, ratio, count, angle, area, and volume, ensuring both diversity and coverage.
>
> 1. We further divided GeoGramBench into second-level subtypes based on the problem-solving objectives, determined via manual annotation. The following table summarizes the detailed statistics, showing that each major category contains a diverse set of subtypes.
>
> 2. Additionally, we calculated fine-grained accuracy on these subtypes for 17 models. For illustration, results from 6 representative models are shown below; additional model results will be included in the revised paper.
>
> 3. Based on these experiments, we also identified several subtype-specific error patterns, which are discussed in detail in the subsequent analysis.
>
> Subtype Statistics
>
> |Category| Angle | Length | Area | Volume | Ratio | Count | Total |
> |:---------|:------:|:---------:|:-----:|:--------:|:------:|:-------:|:-------:|
> | Primitive | 22    | 25     | 26   | 0      | 14    | 15    | 102   |0|
> | Compositional  | 20    | 88     | 89   | 0      | 51    | 31    | 279   |
> | Abstract  | 7     | 20     | 46   | 27     | 4     | 15    | 119   | 0|
>
> Experiments Results
>
>    - Primitive Recognition
> |Model| Avg.|Angle | Length | Area | Ratio | Count |
> |:------|:-------:|:---------:|:------:|:-------:|:--------:|:-----:|
> GPT-o3-mini | 83.49 | 69.57 | 87.50 | 92.24 | 76.47 | 89.47|
> GPT-o1 | 85.92 | 71.15 | 88.39 | 92.67 | 88.23 | 90.13|
> GPT-4o         | 40.02 | 25.48 | 46.43 | 47.84 | 38.23 | 40.13|
> DeepSeek-R1 | 84.68 | 73.86 | 87.50 | 87.98 | 89.29 | 85.83 |
> DeepSeek-v3-0324  | 79.73  | 66.35 | 87.50 | 81.90 | 80.89 | 82.23 |
> QwQ-32B | 85.17 | 75.57 | 86.50 | 82.21 | 90.18 | 90.83
>
>    - Local Relation Composition
> |Model| Avg.|Angle | Length | Area | Ratio | Count |
> |:------|:-------:|:---------:|:------:|:-------:|:--------:|:-----:|
> GPT-o3-mini | 76.10 | 51.96 | 78.91 | 77.37 | 75.42 | 79.55|
> GPT-o1 |  76.12 | 50.00 | 80.63 | 79.60 | 69.07 | 79.55|
> GPT-4o   |  21.36 |  9.78 | 23.00 | 21.93 | 22.03 | 20.45|
> DeepSeek-R1 |  75.13 | 66.8 | 76.85 | 78.93 | 70.10 | 72.98 |
> DeepSeek-v3-0324  | 68.71 | 52.72 | 71.85 | 73.58 | 64.19 | 60.98 |
> QwQ-32B |  73.12 | 51.88 | 77.70 | 79.63 | 65.93 | 69.35|
>    - Global Abstract Integration
> |Model| Avg. | Angle | Length | Area | Volume | Ratio | Count |
> |:------|-------:|---------:|------:|-------:|--------:|-----:|-----:|
> |GPT-o3-mini | 42.67 | 55.36 | 59.56 | 30.14 | 30.60 | 59.38 | 71.32|
> |GPT-o1 | 44.67 | 64.28 | 58.15 | 35.30 | 36.20 | 62.5 | 57.78|
> |GPT-4o| 4.51 | 14.29 | 6.52 | 0.70 | 2.10 | 25.00 | 8.82|
> |DeepSeek-R1| 40.86 | 60.71 | 53.75 | 34.78 | 24.07 | 59.38 | 58.33|
> |DeepSeek-v3-0324  | 28.29 | 51.79 | 45.65 | 16.51 | 18.53 | 56.25 | 41.91|
> |QwQ-32B | 37.92 | 44.64 | 53.23 | 31.25 | 25.46 | 46.88 | 59.17|
>
> Fine-Grained Analysis
> 1. Within the Primitive and Compositional categories, accuracy on angle-related problems is consistently lower than on tasks involving length, area, or ratio.
> 2. In the most complex category, Abstract, volume tasks are the most difficult, with all models showing the lowest accuracy on these problems. These results suggest that reasoning about angles—and especially volumes—remains a major challenge for LLMs, even when they perform reasonably well on other geometric properties.
>
> This pattern suggests that reasoning about angles—and especially volumes—poses significant obstacles for LLMs, even when other geometric properties are handled more effectively. We hypothesize that these challenges reflect the fundamental limitations of current models in handling the abstraction and complex spatial relationships required for reasoning about angles and volumes.

---

> > ### Comment · Reviewer_9Qhh · 2025-08-07
> >
> > Thank you for the detailed response. The new experiments and analyses substantially strengthen the work and resolve my main concerns. I have no further question now.

---

### Official Review · Reviewer_NfAE · 2025-07-02

**Ethics Flags:** Data quality and representativeness
**Rating:** 4
**Confidence:** 3

**Summary:**

This paper introduces a new benchmark GeoGramBench focusing on Program-to-Geometry task. It consists of 500 carefully refined problems organized by a tailored three-level taxonomy that considers geometric complexity rather than mathematical reasoning.  Extensive evaluation on 17 LLMs has indicated the weakness of existing models on geometric reasoning.

**Dataset Code Accessibility:**

Yes

**Ethical Considerations:**

No, there are no or only very minor ethics concerns

**Final Justification:**

Thanks for the clarification from the authors, most of my concerns are resolved. I am still interested in how much gap is due to data scarcity in this newly introduced domain. However, the introduction of this benchmark is still valuable to some degree and thus I am leaned to borderline accept.

**Limitations Weaknesses:**

The major concerns lie in following:

1. While spatial reasoning is an important capability, the proposed program-driven mathematical geometric benchmark targets on a fine-grained domain. Therefore, it is unclear how helpful the proposed benchmark is to the community. Will a better performance on this specific domain boost the general spatial reasoning capability of LLMs? Since most LLMs might not trained on this type of data, the performance gap observed in the experiment might be more of data shortage issue then modeling issue.

2. It seems most of the ground truth in this benchmark are numerical value. Wondering whether the authors measure how closer the model's response to the correct answer and how good the reasoning process is. It is encouraged to compare the thinking process among different thinking models to understand why the error happens.

**Strengths Contributions:**

The introduction of the proposed benchmark is clear and writing is easy to follow. The proposed benchmark is interesting and the experimental results have indicated the weakness of existing LLMs.

---

> ### Author Rebuttal · Authors · 2025-07-30
>
> Thank you for your detailed feedback and thoughtful suggestions. Below, we address your specific comments and provide additional details about our work.
> > While spatial reasoning is an important capability, the proposed program-driven mathematical geometric benchmark targets on a fine-grained domain. Therefore, it is unclear how helpful the proposed benchmark is to the community. Will a better performance on this specific domain boost the general spatial reasoning capability of LLMs?
>
> 1. While Program-to-Geometry is a fine-grained domain, it targets the foundational geometric spatial reasoning ability of LLMs, and is first proposed and clearly defined in this work. (a) Unlike traditional geometric spatial reasoning tasks in the VLM domain—which rely on perception modules to process visual input (e.g., MathVerse, MathVista)—Program-to-Geometry eliminates perceptual factors and directly evaluates the intrinsic geometric representation ability of LLMs. (b) It is a form of formalized geometric figure reasoning that abstracts away from concrete visual attributes, thereby focusing more purely on geometric properties and the inference process itself. (c) Program-to-Geometry specifically targets the internal visualization capability—whether LLMs can internally visualize and reason about geometric concepts—which is essential for autonomous robotics and visual-language-action applications.
> 2. GeoGramBench is the first large-scale, dedicated benchmark specifically designed for Program-to-Geometry tasks. To our best knowledge, existing publicly available datasets in this domain contain fewer than 50 problems, whereas GeoGramBench provides 500 carefully curated and diverse problems. This substantial increase in scale and diversity fills a significant gap, laying a strong foundation for future research in symbolic-to-spatial geometric reasoning.
>
> In summary, although Program-to-Geometry is a fine-grained task, its focus on intrinsic geometric reasoning fills a key gap. GeoGramBench provides the first large-scale resource for systematic evaluation in this area, offering a valuable tool for advancing geometric and spatial intelligence research.
>
> > Since most LLMs might not trained on this type of data, the performance gap observed in the experiment might be more of data shortage issue then modeling issue.
>
> Thank you for your valuable comment. To address this question, we conducted ablation experiments with SFT using domain-specific data, in order to determine whether the observed performance gap is primarily due to training data scarcity or to intrinsic limitations in LLM geometric representation. For details on the SFT setup and results, please refer to our response to Reviewer gGrj. Here, we summarize the main findings:
>
> 1. Domain data scarcity is a factor. Compared to the baseline, adding just 100 domain-specific training samples for fine-tuning leads to significant improvement across all three difficulty levels, increasing average accuracy by 3.02 percentage points. This indicates that lack of exposure to this type of data—such as program code, geometric representations, and specific data formats—does contribute to the models’ underperformance.
>
> 2. Scaling data yields limited returns. When we increased the number of training samples by two- and three-fold, the accuracy improvements became marginal. Even when we tripled the number of domain samples, the accuracy improvement was less than 0.6 percentage points (+3.02 vs. +3.60), highlighting the limited returns from simply scaling up domain-specific data. This plateau indicates that while domain data scarcity accounts for some of the performance gap, it is not the primary bottleneck. Instead, the main limitation appears to be the model’s weak intrinsic geometric spatial representation ability.
>
> 3. Persistent difficulty for complex tasks. Even after adding 300 in-domain samples, the models still struggled on the most challenging “Abstract” subset, achieving less than 20% accuracy, and averaged only around 40% overall. This further highlights the fundamental challenge in the models’ internal capacity for geometric reasoning.
>
> In summary, while a shortage of domain-specific data does affect performance, the primary limitation lies in the models’ inherently weak symbolic-to-spatial reasoning abilities.
>
> > It seems most of the ground truth in this benchmark are numerical value. Wondering whether the authors measure how closer the model's response to the correct answer and how good the reasoning process is.
>
> 1. All the ground truth in GeoGramBench are numerical, similar to benchmarks like AIME24 and MATH500 Bench. For answer accuracy, we follow established community practices and utilize reference implementations from robust, open-source libraries such as *EleutherAI/lm-evaluation-harness* and *huggingface/Math-Verify*, which standardize number formatting (including LaTeX representations), strip units if present, and compare the processed values to determine correctness.
>
> 2. Like most MATH-style benchmark datasets (e.g., AIME24/25, MATH500, GSM8K), GeoGramBench evaluates only answer correctness without assigning explicit scores to the quality of the reasoning process. While evaluating the reasoning process is crucial for deeper insight into LLM characteristics, it remains highly challenging as it requires rigorous assessment of symbolic logic and multi-step reasoning. Developing automated quantitative protocols for this purpose is left for future work, and leveraging advanced LLMs as automated evaluators may be a promising direction.
> 3. We provide a qualitative analysis of model reasoning, including extensive failure case studies presented in Section 6 and Appendix E. This analysis focuses on three aspects—Construction Evidence, Compositional Completeness, and Chain-of-Thought (CoT) Reasoning—each illustrated with detailed examples and supporting arguments.
>
> > It is encouraged to compare the thinking process among different thinking models to understand why the error happens.
>
> Thank you for this insightful suggestion. Through extensive qualitative analysis of model responses—conducted via manual examination of failure cases—we identified several common failure patterns across models:
>
> 1. Preference for algebraic problem-solving: The models heavily rely on algebraic methods, such as using coordinates to directly compute quantities like segment lengths, rather than employing geometric constructions ("*Since I can't see the diagram, maybe I should proceed with the information given...BC is from (4,0) to (8,0), which is 4 units, so length 4*").
> 2. Limited ability to utilize auxiliary lines: In many cases, introducing auxiliary lines would offer a much simpler solution, yet the models consistently avoid this approach and instead pursue more complex strategies, which often result in failure.
> 3. Difficulty with spatial directions: The models frequently struggle with instructions involving spatial orientation, such as clockwise, counterclockwise, or references to vertical and horizontal directions ("*So the inner arc is fr
> om a/2 to c/2, clockwise?...which would be the other direction compared to the inner counterclockwise path before*" ).
> 4. Confusion in mapping geometric relationships: The models often display confusion when relating symbolic elements to geometric components, such as associating symbols for angles or vertex labels with the corresponding diagram elements. This likely stems from a lack of strong spatial understanding or an internal world model, making it difficult to map abstract symbols to concrete geometric entities ("*Maybe E is on one trisection and F on another? Hmm, this is getting a bit confusing without seeing the actual figure...Maybe it's labeling an angle at point B? But angle at B is between A, B, and C*").
>
> These observations yield useful insights into model weaknesses and can inform more systematic future comparisons of reasoning processes. They also provide a foundation for more rigorous and quantitative analyses of the thinking processes exhibited by different kinds of models.

---

### Official Review · Reviewer_U3SF · 2025-07-02

**Rating:** 5
**Confidence:** 4

**Summary:**

The paper first defines “program-to-geometry” tasks from interpretation of procedural drawing code
to the downstream geometric reasoning it enables, which motivates the need for GeoGramBench. GeoGramBench introduces a curated set of 500-problem with GPT-4o based filtering and human refinements. It uses a new 3-level taxonomy (primitive recognition, local relation, global integration) and rigorously filters problems to avoid answer leakage. Across 17 leading LLMs, even the best models achieve <50% accuracy on the hardest problems, exposing deep limitations in symbolic-to-geometric reasoning among current existing SOTA LLMs.

**Dataset Code Accessibility:**

Yes

**Ethical Considerations:**

No, there are no or only very minor ethics concerns

**Final Justification:**

The authors have provided clarifications to my concerns. I believe this paper's dataset contributions (in encouraging research in the “program-to-geometry” direction) overshadow its weakness.

**Limitations Weaknesses:**

- GeoGramBench only covers 2D procedural code; does not test multimodal (image+code) or real-world 3D geometric tasks.
- data effectiveness: SFT experiments showing the effectiveness of GeoGramBench in improving LLMs' performance on “Program-to-Geometry” tasks, also potentially its generalizability (to other spatial reasoning tasks) would be useful.

**Strengths Contributions:**

- First benchmark specifically for “Program-to-Geometry” tasks in LLMs.
- GeoGramBench employs extensive automated and human filtering to eliminate answer leakage, pretraining contamination, and ambiguity.
- dataset structure and quality: GeoGramBench organizes problems by geometric complexity: primitive recognition, local relation, and global abstract integration from diverse math datasets (NuminaMath, HARP, Omni-MATH, AIME24, Mathverse, MATH-500).
- evaluation: GeoGramBench benchmarks 17 open- and closed-source models, with detailed behavior and error analysis.
- Code and data are open-sourced.

---

> ### Author Rebuttal · Authors · 2025-07-30
>
> Thank you for your encouraging review and valuable feedback! We are especially appreciative that you recognized the value and potential impact of our contribution. The following provides additional clarifications and experiments incorporated into the paper based on your feedback. Please let us know if there are any further improvements we can make.
>
> > GeoGramBench only covers 2D procedural code; does not test multimodal (image+code) or real-world 3D geometric tasks.
>
> Indeed, extending GeoGramBench from 2D to multimodal and real-world 3D geometric tasks is a valuable direction for future work. The core contribution of this paper lies in:
> 1.  We clearly define the value and task characteristics of “Program-to-Geometry.” This task probes the geometric spatial reasoning abilities of LLMs without relying on perception modules, thus removing confounding perceptual factors. By adopting a formalized geometric representation, the evaluation is more focused and accurate.
> 2. Accordingly, we present GeoGramBench as a foundational benchmark for this direction, aiming to provide the community with a solid research base for advancing studies in multimodal and 3D scenarios.
>
> > data effectiveness: SFT experiments showing the effectiveness of GeoGramBench in improving LLMs' performance on “Program-to-Geometry” tasks, also potentially its generalizability (to other spatial reasoning tasks) would be useful.
>
> 1. We have conducted SFT experiments using GeoGramBench, as discussed in detail in our response to Reviewer gGrj. The results clearly demonstrate the effectiveness of this dataset for improving LLM performance on Program-to-Geometry tasks:
>     - Significant gains with limited in-domain data: Adding as few as 50 in-domain training examples from GeoGramBench increased average accuracy by 1.62 points, while 100 examples yielded an improvement of 3.02 points on Program-to-Geometry tasks. This demonstrates the high data efficiency of GeoGramBench and shows that even a small amount of targeted task-specific data can meaningfully address some deficiencies in spatial understanding due to data scarcity.
>     - Further improvements with more data, but an upper bound exists: Increasing the number of in-domain examples to 300 further boosted the improvement to 3.60 points. Although the additional gains were smaller, this trend is expected: as the model’s exposure to relevant data increases, performance plateaus due to the intrinsic limitations of current LLMs in geometric representation and spatial reasoning. These results indicate that while task-specific data is effective in bridging some capability gaps, it cannot fully compensate for fundamental weaknesses in the model’s underlying geometric reasoning ability.
>
> 2. Leveraging GeoGramBench data for broader generalization to a wide range of spatial understanding and generation tasks would be highly impactful. We see evaluating such transfer effects, both within and beyond the Program-to-Geometry domain, as a meaningful extension of this work.
>
> Finally, we will incorporate the data effectiveness experiments into the paper, and will explicitly discuss the extension to multimodal tasks, real-world 3D geometric tasks, and generalizability to other spatial reasoning tasks in the Future Work section of the paper.

---

> > ### Comment · Reviewer_U3SF · 2025-08-08
> >
> > Thank you for your clarifications. My concerns are addressed and I will maintain my recommandation.

---

### Note · Authors · 2025-08-12

We sincerely thank all reviewers for their detailed and constructive feedback, and we are pleased that most of their concerns have been thoroughly addressed, with no further issues raised. Through comprehensive clarifications and targeted additional experiments, we have significantly strengthened both the content and empirical validation of our paper. The relevant revisions and additional experimental results will be incorporated into the final version of the paper.

- For Reviewer U3SF’s concern regarding data effectiveness, we conducted SFT experiments demonstrating that GeoGramBench substantially improves LLM performance on Program-to-Geometry tasks with limited in-domain data. Reviewer U3SF confirmed that their concerns have been resolved and maintained their recommendation.

- Regarding Reviewer NfAE’s question about whether the observed performance gap stems from data scarcity or intrinsic model limitations, our ablation studies show that although data scarcity partly affects performance, scaling data yields diminishing returns. The primary bottleneck lies in the models’ inherent geometric spatial reasoning capabilities. Reviewer NfAE appreciated our detailed analysis, including the qualitative dissection of failure modes, and indicated that most concerns have been addressed.

- To address Reviewer 9Qhh’s comments on CoT prompting strategies and the granularity of error analysis, we supplemented our work with quantitative ablation experiments on CoT methods and conducted a manual, fine-grained classification and statistical analysis of problem types. This detailed breakdown provides valuable insights into the specific subtask challenges faced by the models. Reviewer 9Qhh acknowledged these enhancements and confirmed that their concerns have been resolved.

- For Reviewer gGrj’s inquiry on fine-tuning task-specific models and larger LLMs, we performed systematic data ablation experiments showing that while lack of task-specific data reduces initial performance, the essential limitation remains the models’ intrinsic geometric reasoning ability. Reviewer gGrj has not yet provided further feedback.

In summary, our comprehensive responses and rigorous experiments have satisfactorily addressed the key issues raised by most reviewers. We hope this strengthens the paper’s evaluation and thank all reviewers and Area Chairs for their valuable time and effort.

---

### Decision · Program_Chairs · 2025-09-18

**Decision:**

Reject

**Comment:**

This paper introduces GeoGramBench, a benchmark for evaluating the geometric program reasoning of LLMs. It claims that current models struggle with this task, showing a fundamental weakness in translating procedural code into internal spatial representations. The findings confirm this, demonstrating that even advanced models achieve low accuracy on complex geometric problems. The paper's main strength is its introduction of a novel and important task, "Program-to-Geometry," supported by a meticulously curated and reproducible dataset. It also provides a thorough evaluation and analysis of LLM performance. The initial weaknesses, a narrow focus and missing analyses, were fully addressed in the rebuttal. The authors provided comprehensive responses that resolved all reviewer concerns, particularly by adding new experiments that confirmed the performance gap is due to a fundamental modeling issue, not just data scarcity. The paper's contribution is a valuable new resource for the research community. In sum, I recommend acceptance.

===== FINAL UPDATE FROM DB Track PCs ====

The final decision for this paper has been taken by the program chairs after consultation with the SACs. All Senior Area Chairs have ranked papers according to the feedback from the AC during the review process. We decided to leave the original meta-review to reflect the opinion of the AC in light of the initial discussions with reviewers and SAC.